# Simultaneous Analysis of the p16 Gene and Protein in Canine Lymphoma Cells and Their Correlation with pRb Phosphorylation

**DOI:** 10.3390/vetsci9080393

**Published:** 2022-07-29

**Authors:** Leni Maylina, Satoshi Kambayashi, Kenji Baba, Masaru Okuda

**Affiliations:** 1Laboratory of Veterinary Internal Medicine, Joint Graduate School of Veterinary Medicine, Yamaguchi University, 1677-1 Yoshida, Yamaguchi 753-8515, Japan; leni_maylina@apps.ipb.ac.id (L.M.); s-kam@yamaguchi-u.ac.jp (S.K.); kbaba@yamaguchi-u.ac.jp (K.B.); 2Division of Veterinary Internal Medicine, Department of Clinic, Reproduction and Pathology, School of Veterinary Medicine and Biomedical Sciences, IPB University, Jl. Agatis, Bogor 16680, Indonesia; 3Laboratory of Veterinary Internal Medicine, Joint Faculty of Veterinary Medicine Yamaguchi University, 1677-1 Yoshida, Yamaguchi 753-8515, Japan

**Keywords:** canine lymphoma, methylation, p16 (CDKN2A), pRb phosphorylation (pRb-P)

## Abstract

**Simple Summary:**

Lymphoma is one of the most frequently diagnosed malignancies in dogs. The most common epigenetic alteration is gene methylation. Methylation of the p16 gene leads to decreased expression of its protein. The p16 protein inhibits the activity of cyclin-dependent kinase, as a negative control of the cell cycle to prevent phosphorylation of the retinoblastoma (pRb) protein. The methylation of the p16 gene has been reported in canine lymphomas, however, p16 protein expression has not been examined in previous studies. In this study, the gene and protein expression of p16, and phosphorylation of pRb, were examined simultaneously in canine lymphoma/leukemia cell lines treated with or without a demethylation drug in vitro. We identified the hypermethylation of the p16 gene, the decreased expression of p16 protein and the hyperphosphorylation of pRb in four out of eight cell lines. Furthermore, we revealed that the expression of the p16 protein was more stable than that of the p16 gene and more closely related to the phosphorylation of pRb. In conclusion, the p16 protein expression is suggested as a promising biomarker for canine lymphoma cells, and the p16–pRb pathway could be a target for the better treatment of canine lymphomas.

**Abstract:**

Cyclin-dependent kinase inhibitor p16 (CDKN2A) primarily functions as a negative regulator of the retinoblastoma protein (pRb) pathway to prevent pRb phosphorylation, thus playing a critical role in cell cycle arrest. In canine lymphoma cells, methylation due to inactivation of the p16 gene has been reported. However, its protein expression has not been examined in previous studies. In our in vitro study, the gene and protein expression of p16 and phosphorylated pRb were examined simultaneously in eight canine lymphoma and leukemia cell lines (17-71, CLBL-1, GL-1, CLC, CLGL-90, Ema, Nody-1, and UL-1). Methylation of the p16 gene was also explored using the demethylation drug 5-Aza-2′-deoxycytidine (5-Aza). After 5-Aza treatment, p16 gene and protein expression increased and pRb phosphorylation decreased, suggesting that both hypermethylation of the p16 gene and pRb hyperphosphorylation occurred in four out of eight cell lines (CLBL-1, CLC, Nody-1, and UL-1). Moreover, the estimation of p16’s protein expression was better than that of p16’s mRNA expression because the expression of the protein was more stable than those of the gene, and highly related to the phosphorylation of pRb. These results revealed that p16’s protein expression could be a promising biomarker for canine lymphoma cells.

## 1. Introduction

Lymphoma is one of the most frequently diagnosed malignancies in dogs and is comparable to human lymphoma [1,2]. The incidence rate of canine lymphomas is estimated at 20–100 cases per 100,000 dogs [2]. Life expectancy varies depending on the classification, such as grade or immunophenotype [3]. In a study of 608 cases of canine malignant lymphoma, 24.5% and 75.5% were classified as low- and high-grade lymphomas, respectively [4]. Low-grade lymphomas usually show an indolent nature and tend to survive longer than the high-grade lymphomas. Canine B-cell lymphomas have a high prevalence (97/123; 79.9%) [5]. Moreover, molecular immunophenotypes present in the dogs under study were B-cell lymphomas (753/1226; 61.4%) and T-cell lymphomas (473/1226; 38.6%) [6]. Remission and survival times in high-grade T-cell lymphomas have been shown to be shorter than those in high-grade B-cell lymphomas [1,2].

Many forms of human lymphoma have specific genetic abnormalities. These abnormalities can be used as diagnostic and prognostic factors [7]. The translocation of t(8;14) (q24;q32), which results in the rearrangement of the MYC proto-oncogene with an immunoglobulin heavy chain, is an important genetic characteristic of Burkitt lymphoma [8,9,10]. In follicular lymphoma (FL), a translocation of t(14;18)(q32;q21), which results in an augmented expression of the *BCL2* gene, is often used for diagnostic purposes [10,11,12]. Overexpression of the cyclin D1 protein, a regulator of the early phases of the cell cycle through inactivation of the tumor suppressor retinoblastoma protein (pRb), plays an important role in the pathogenesis of mantle cell lymphoma (MCL) [10,13]. Moreover, aberrant epigenetic alterations, including the regulation of the expression of genes and signal transduction, play an important role in the pathogenesis and development of lymphoma. The most common epigenetic alterations are DNA methylation and histone modification [14]. Although similar studies are currently being conducted on canine lymphomas [15], the information on their molecular biology is still limited.

Cyclin-dependent kinase inhibitor p16 (CDKN2A) primarily functions as a negative regulator of cell cycle progression via the pRb pathway. The p16 protein inhibits the activity of cyclin-dependent kinase (CDK), prevents the phosphorylation of pRb, and thus plays a critical role in cell cycle arrest [16]. In human lymphomas, several studies have demonstrated that hypermethylation of the p16 promoter region leads to the loss of p16 gene expression [17]. In human Burkitt lymphoma, p16 methylation was detectable in 72% of cases, whereas p16 promoter methylation was detected in 80% of patients with stage III/IV [18]. In human diffuse large B-cell lymphoma (DLBCL), p16 is one of the most frequently deleted driver genes [19], and its genetic alterations are associated with significantly poorer survival [20].

Inactivation of p16 has been reported in canine lymphomas. Decreased expression of the p16 gene has been reported in the CLBL-1, GL-1, Nody-1, Ema, and UL-1 canine lymphoma cell lines, and hypermethylation of the p16 gene has also been identified in the CLBL-1, GL-1, and UL-1 cell lines [21,22]. Moreover, decreased expression of the p16 gene (B-cells: 32/49, 65%; T-cells 9/13, 69%), as well as its hypermethylation (B-cells: 13/55, 24%; T-cells 7/13, 54%) have been reported in canine lymphoma cells obtained from naturally occurring clinical cases [23]. In addition, in canine high-grade T-cell non-Hodgkin lymphoma (NHL) cases, deletion of the p16 gene and pRb phosphorylation reached 100% [16], suggesting that the p16 gene deletion and pRb phosphorylation are prognostically valuable parameters for canine NHL. Moreover, in canine T-cell lymphomas, p16 inactivation through loss of chromosome 11, in which the p16 gene is located, or deletion of the p16 gene were found to be correlated with poor prognosis [24]. However, in these previous studies, p16 protein expression was not examined, as the appropriate antibody that can detect canine p16 protein was not identified until 2018 [25]. To the best of our knowledge, simultaneous analyses of the p16 gene, its protein expression, and their correlation with Rb protein phosphorylation have not been performed in canine cancers.

In this study, to clarify the direct relationship between the inactivation of the p16 gene and its protein expression, as well as pRb activation, we (1) identified whether decreased p16 gene expression was related to the absence of that protein expression; (2) determine if the decrease in p16 gene and protein expression was associated with the methylation of that gene; (3) identified if any decrease was related to the hyperphosphorylation of the Rb protein; and (4) determined if the removal of the methylation of the p16 gene was related to the decrease in phosphorylation of the Rb protein.

## 2. Materials and Methods

### 2.1. Cells

In the present study, we used eight canine lymphoma and leukemia cell lines derived from dogs with naturally occurring lymphoid malignancies. These included the following B-cell lines: 17-71 (multicentric B-cell lymphoma), CLBL-1 (multicentric B-cell lymphoma), and GL-1 [B-cell acute lymphoblastic leukemia (ALL)] [26,27,28]; and T-cell lines: CLC (gastrointestinal T-cell lymphoma), CLGL-90 (chronic large granular lymphocytic T-cell leukemia), Ema (mediastinal T-cell lymphoma), Nody-1 (alimentary T-cell lymphoma), and UL-1 (renal T-cell lymphoma) [29,30,31,32]. All canine lymphoma and leukemia cell lines were maintained in a complete medium [RPMI-1640 (FUJIFILM Wako Pure Chemical Corporation, Tokyo, Japan) containing 10% fetal bovine serum (FBS) and 1% penicillin-streptomycin (Nacalai Tesque, Kyoto, Japan)]. We used HEK293T human cell line as a positive control of p16 protein expression for western blot analysis (Appendix A). HEK293T human cell line was maintained in a complete medium [D-MEM (high glucose) with L-glutamine and phenol red (FUJIFILM Wako Pure Chemical Corporation) containing 10% FBS and 1% penicillin-streptomycin (Nacalai Tesque)]. Live cells equal to or greater than 90% are used to initiate cell culture. Furthermore, we used *Mycoplasma*-free cells with passage times under thirty. All the cells were maintained at 37 °C in humidified air containing 5% CO_2_.

### 2.2. Treatment of Cell Lines with 5-aza-2-Deoxycytidine

A demethylation drug, 5-aza-2′-deoxycytidine (5-Aza, Sigma–Aldrich, Saint Louis, MO, USA), which causes DNA demethylation, was dissolved in distilled water at a concentration of 43.8 mM and stored at −20 °C until required. The treatment of the canine lymphoma cell lines with 5-Aza according to the indicated doses (CLC and Nody-1 cell lines, 0.25 µM; 17-71, CLBL-1, GL-1, CLGL-90, Ema, and UL-1 cell lines, 0.5 µM) was determined based on our preliminary experiments (Appendix A). The cells were cultured with or without 5-Aza for 72 h. The medium with or without 5-Aza was changed every 24 h. At the end of the culture period, total RNA and protein amounts were extracted from the cell lines, as described below.

### 2.3. Real-Time Polymerase Chain Reaction (PCR)

Total RNA was extracted using the NucleoSpin RNA Plus Kit (TakaraBio, Shiga, Japan). Complementary DNA (cDNAs) was synthesized using the ReverTra Ace qPCR RT Kit (TOYOBO, Osaka, Japan). Single-strand cDNAs were subjected to real-time PCR amplification using the THUNDERBIRD^TM^ SYBR^TM^ qPCR Mix (TOYOBO) according to the manufacturer’s instructions. The quantity of total RNA and cDNA was calculated using NanoDrop 2000 Spectrophotometer (Thermo Fisher Scientific, Waltham, MA, USA), with the quality assessment of nucleic acid based on the absorbance waveform. The RNA and cDNA were stored in the freezer at −30 °C before use. The p16 mRNA expression level was measured by real-time PCR using a CFX96 Touch Real-Time PCR Detection System (Bio-Rad, Hercules, CA, USA) with p16 primers (p16F and p16R; amplicon length 95 bp, Table 1) [21]. Ribosomal protein L32 (RPL32) was used as an endogenous control (RPL32F and RPL32R; amplicon length 100 bp, Table 1) [33]. The cycling protocol was as follows: denaturation step at 95 °C for 30 s; 45 cycles of denaturation at 95 °C for 5 s, annealing at 60 °C for 10 s and extension at 72 °C for 10 s. PCR amplicons were electrophoresed on a 3% agarose gel. The comparative cycle threshold (Ct) method was used to quantify the p16 transcript levels. ΔCt was determined by subtracting the Ct value of RPL32 from that of p16 mRNA. The levels of p16 mRNA relative to RPL32 were calculated as 2^−ΔCt^. All samples were evaluated in three replicates of the triplicate assay.

### 2.4. Western Blot Analysis

Briefly, canine lymphoma cells were lysed in SDS lysis buffer with protease and phosphatase inhibitors (Thermo Fisher Scientific) and kept on ice or stored at −80 °C if not used immediately. Cell lysates were electrophoresed on 6% or 12% SDS polyacrylamide gels at 2 A per gel for one hour. The lysates were then transferred onto a PVDF membrane (Merck Millipore, Billerica, MA, USA) at 100 V for one hour. The PVDF membrane was washed three times in Tris-buffered saline containing 0.1% Tween-20 (TBST), blocked for one hour with 5% nonfat milk/TBST (blocking buffer), and subsequently washed three more times with TBST. The PVDF membrane was then incubated overnight at 4 °C with mouse monoclonal anti-p16 (1:500, F-8; Santa Cruz Biotechnology, Dallas, TX, USA) [25], rabbit monoclonal anti-phospho-pRb (1:1000, phospho-T826; Abcam, Fremont, CA, USA), or mouse monoclonal anti-pRb (1:1000, G3-248; BD Pharmingen, San Diego, CA, USA) [16] diluted in 5% nonfat milk/TBST (antibody dilution). Mouse monoclonal anti-β-actin (1:5000, AC-15; Sigma–Aldrich) diluted in 0.5% non-fat milk/TBST was used as the endogenous control. The antibodies used are summarized in Table 2. The membranes were washed three times for 10 min each and incubated with HRP-conjugated goat anti-mouse or anti-rabbit IgG antibody (Santa Cruz Biotechnology) for one hour at room temperature. Finally, membranes were washed three times for 10 min, then incubated for 5 min with SuperSignal^TM^ West Pico PLUS Chemiluminescent Substrate reagent (Thermo Fisher Scientific) and imaged using an AMERSHAM ImageQuant 800 (GE Healthcare Bio-Sciences AB, Uppsala, Sweden).

### 2.5. Statistical Methods

The real-time PCR values of p16 mRNA expression were assessed using CFX Maestro software (Bio-Rad). p16 protein expression was quantified using ImageJ software [34]. The p16 mRNA and protein expression were analyzed using nonparametric test by independent-samples Kruskal–Wallis test, for which significance values have been adjusted by Dunn–Bonferroni correction for multiple testing. Furthermore, the p16 mRNA and protein expression values with or without 5-Aza treatment were analyzed using a sample-paired *t*-test. Statistical significance was set at *p* < 0.05.

## 3. Results

### 3.1. The Expression of p16 mRNA Was Correlated with p16 Protein Expressions

To analyze the expression correlation between the p16 gene and protein, we examined the p16 gene and protein expressions in eight canine lymphoma and leukemia cell lines (B-cells: 17-71, CLBL-1, and GL-1; T-cells: CLC, CLGL-90, Ema, Nody-1, and UL-1) using real-time PCR and western blot analysis, respectively (Figure 1).

The relative expression levels of p16 mRNA in the CLBL-1, CLC, CLGL-90, Ema, Nody-1, and UL-1 cell lines were between 1.0 × 10^−5^ and 1.0 × 10^−4^. In contrast, those of the 17-71 and GL-1 cell lines were between 1.0 × 10^−3^ and 1.0 × 10^−2^ (Figure 1a), showing that the expression levels of p16 mRNA in the CLBL-1 B-cell line and in the CLC, CLGL-90, Ema, Nody-1, and UL-1 T-cell lines were significantly lower than those in the 17-71 and GL-1 B-cell lines.

Furthermore, Figure 1b,c show the expression levels of the p16 protein observed in the 17-71 and GL-1 B-cell lines, where the p16 mRNA expression levels were significantly high. In contrast, we did not observe p16 protein expression in the CLBL-1 B-cell line or in the CLC, CLGL-90, Ema, Nody-1, and UL-1 T-cell lines, where the expression levels of the p16 gene were low, indicating that the expression levels of the p16 gene and protein are highly correlated.

### 3.2. The 5-Aza Treatment Induced the Expressions of the p16 Gene and Protein; and Decreased pRb Phosphorylation in Some Canine Lymphoma and Leukemia Cell Lines

To determine whether the decreased expression of the p16 gene and protein was associated with the methylation of that gene, the two groups of cell lines were treated with 5-Aza and compared with groups without it using real-time PCR and western blot analysis.

Figure 2 shows the expression of p16 gene in canine lymphoma cells treated with or without 5-Aza. After 5-Aza treatment, the expression levels of p16 mRNA were significantly increased in B-cells (17-71, CLBL-1, and GL-1) and T-cells (CLC, Nody-1, and UL-1). In contrast, p16 gene expression was not altered in the CLGL-90 and Ema cell lines after 5-Aza treatment.

Figure 3 and Figure 4 show the expression of the p16 protein, the phosphorylation of pRb, and total pRb in canine lymphoma cells with or without 5-Aza. In addition to the 17-71 and GL-1 cell lines, p16 protein expression was observed in the CLBL-1, CLC, Nody-1, and UL-1 cells treated with 5-Aza. In contrast, p16 gene and protein expression levels were not altered and still showed low expression in the CLGL-90 and Ema cell lines after 5-Aza treatment (Figure 2 and Figure 4). The results indicated that p16 protein expression was suppressed by methylation of the p16 gene in the CLBL-1, CLC, Nody-1, and UL-1 cell lines; but not in the CLGL-90 and Ema cell lines.

The expression levels of the p16 mRNA in the 17-71 and CLBL-1 cell lines without the 5-Aza treatment, as shown in Figure 1a and Figure 2, were different (for example in 17-71: 8.07 × 10^−3^ ± 4.22 × 10^−3^ vs. 6.53 × 10^−4^ ± 8.83 × 10^−5^; CLBL-1: 3.64 × 10^−5^ ± 2.85 × 10^−5^ vs. 3.77 × 10^−4^ ± 1.88 × 10^−4^). In contrast, the expression levels of the p16 protein in the 17-71 and CLBL-1 cell lines without the 5-Aza treatment, as shown in Figure 1c and Figure 3b, were quite similar (for example in 17-71: 0.49 ± 0.04 vs. 0.87 ± 0.07; CLBL-1: 0 vs. 0).

To examine whether the decreased p16 gene and protein expression was related to the hyperphosphorylation of pRb, we explored the phosphorylation of pRb by western blot analysis using cell lines treated with or without 5-Aza. After 5-Aza treatment, the phosphorylation level of pRb decreased in the CLBL-1 B-cell line, in which p16 protein expression was induced (Figure 3). In contrast, the pRb phosphorylation was not identified in the 17-71 and GL-1 cells treated with and without 5-Aza, in which the expression of the p16 protein was high regardless of the 5-Aza treatment.

Figure 4 shows the phosphorylation of pRb in the T-cell lines. After demethylation treatment, pRb phosphorylation was significantly decreased in the CLC, Nody-1, and UL-1 cell lines, and p16 gene and protein expression were increased (Figure 4c). In contrast, the CLGL-90 cell line, which had low expression of the p16 protein, still significantly exhibited phosphorylation of pRb after 5-Aza treatment. Moreover, the phosphorylation of pRb was not altered in the Ema cell line.

The total pRb expression was significantly increased in the CLBL-1 B-cell line (Figure 3d) and the Nody-1 T-cell lines (Figure 4d) after demethylation treatment using 5-Aza. In contrast, the total pRb expression was not altered in the 17-71, GL-1, CLC, CLGL-90, Ema, and UL-1 cell lines treated with 5-Aza.

## 4. Discussion

Inactivation of the p16 gene has been reported in canine lymphoma cells [16,21,22,23,24]. However, the expression of the p16 protein has not been examined in previous studies. The present study clarified the direct relationship between inactivation of the p16 gene and decreased protein expression; and pRb phosphorylation in canine lymphoma cells. To the best of our knowledge, this is the first report that methylation of the p16 gene results in decreased expression of the p16 protein and pRb phosphorylation in canine tumor cells.

In previous study of canine lymphoma cells, low expression of the p16 gene was reported in the CLBL-1, GL-1, Nody-1, Ema, and UL-1 cell lines, and hypermethylation of the p16 gene was observed in the CLBL-1, GL-1, and UL-1 cell lines [21,22]. Furthermore, in a different report, deletion of the p16 gene and phosphorylation of pRb was confirmed in 100% of canine high-grade T-cell non-Hodgkin lymphoma (NHL) cases [16].

The p16 gene inactivation by deletions, mutations, and hypermethylation has been reported to be associated with aggressive variants in human NHLs [35]. In humans, loss of the p16 gene is significantly correlated with the stage of human DLBCL [36], and p16 promoter methylation was detected in 80% of patients with stage III/IV Burkitt’s lymphomas [18]. Furthermore, in human DLBCLs, the loss of the p16 (CDKN2A) and TP53 genes was observed in 35% and 8% of patients, respectively, and was significantly associated with shorter survival after rituximab, cyclophosphamide, doxorubicin, vincristine, and prednisone (R-CHOP) treatment [37]. Based on the above, the examination of the inactivation of p16 gene and protein expression, as well as hyperphosphorylation of Rb proteins, may be further valuable parameters for prognostic decision-making and treatment biomarkers for naturally-occurring canine lymphoma cases.

Table 3 summarizes the p16 gene and protein expression analyses of the p16 gene and protein, and pRb phosphorylation in canine lymphoma cell lines treated with or without 5-Aza. When p16 gene expression was low, the expression of p16 protein was not observed. After 5-Aza treatment, p16 gene and protein expression was increased in the CLBL-1, CLC, Nody-1, and UL-1 cell lines. In addition, the p16 protein expression stably expressing in the 17-71 and GL-1 cells with and without 5-Aza treatment. The phosphorylation of pRb was decreased in the CLBL-1, CLC, Nody-1, and UL-1 cell lines treated with 5-Aza, where the p16 gene and protein expression levels were increased. These results suggest that methylation of the p16 gene and hyperphosphorylation of pRb occurred in the CLBL-1, CLC, Nody-1, and UL-1 cell lines. In contrast, the CLGL-90 and Ema cell lines treated with 5-Aza showed low expression levels of the p16 gene and protein; and phosphorylation of pRb. In contrast, the cell lines in which the p16 gene and protein expression levels were high, 17-71 and GL-1 cells were not related to methylation of the p16 gene and hyperphosphorylation of pRb, suggesting that a mechanism other than inactivation p16 and phosphorylation of pRb is important in these cell lines.

In the present study, we observed low expression of the p16 gene and protein in the CLBL-1, CLC, CLGL-90, Ema, Nody-1, and UL-1 cell lines. We also confirmed higher expression of the p16 gene and protein in the 17-71 and GL-1 cell lines. In contrast, in previous studies, lower expression and hypermethylation of the p16 gene were identified in the GL-1 cell line [21,22]. Although we could not identify the reason for this discrepancy, characteristic alterations may have occurred in the GL-1 cell line during culture at different laboratories.

Deletion of p16 gene and hyperphosphorylation of pRb has been reported to reach 100% in canine high-grade T-cell non-Hodgkin lymphoma (NHL) cases [16]. Our study showed that phosphorylation of pRb may have occurred in 33.3% of the B-cell lines (CLBL-1; 1/3) and in 100% of the T-cell lines (CLC, CLGL-90, Ema, Nody-1, and UL-1; 5/5). However, 40% of the T-cell lines (CLGL-90 and Ema; 2/5), which had low expression of the p16 gene and protein, were not altered after demethylation. In addition, after 5-Aza treatment, pRb phosphorylation was not altered in these cell lines. We could not determine p16 gene methylation in the CLGL-90 and Ema cell lines; aberrations of the p16 gene, including deletions, may have occurred in these cells. Inactivation of p16 gene due to deletion and methylation has been reported in canine lymphomas [16,21,22,23]. The deletion status of the p16 gene was not confirmed in our study; and should be determined in the future.

The expression analysis of the p16 gene by the real-time PCR was not stable, but that of the p16 protein by western blot analysis was shown to be steadily expressed. Furthermore, the standard deviations (SDs) of the p16 protein expression levels were smaller than those of the p16 gene expression levels. Moreover, pRb phosphorylation was found to be highly related to p16 protein expression. When p16 protein levels were low, the phosphorylation of pRb was observed. In contrast, when the p16 protein levels were high, pRb phosphorylation decreased. Therefore, our study revealed that it would be better to examine p16 protein expression rather than p16 gene expression to detect the pRb pathway in canine lymphoma cells. These results suggest that the expression of the p16 protein could be a promising biomarker.

## 5. Conclusions

The p16 gene and protein expression was low in six out of eight (6/8) canine lymphoma cell lines, in which four out of six (4/6) cell lines (CLBL-1, CLC, Nody-1, and UL-1) may have hypermethylation of the p16 gene, decreased expression of the p16 protein, and hyperphosphorylated pRb. The estimation of p16 protein expression was better than that of p16 mRNA expression; because the expression of the protein was more stable than those of the gene, and highly related with the phosphorylation of pRb. These results indicate that p16 protein expression could be a promising biomarker in canine lymphoma cells and might be a treatment target in the future.

## Figures and Tables

**Figure 1 vetsci-09-00393-f001:**
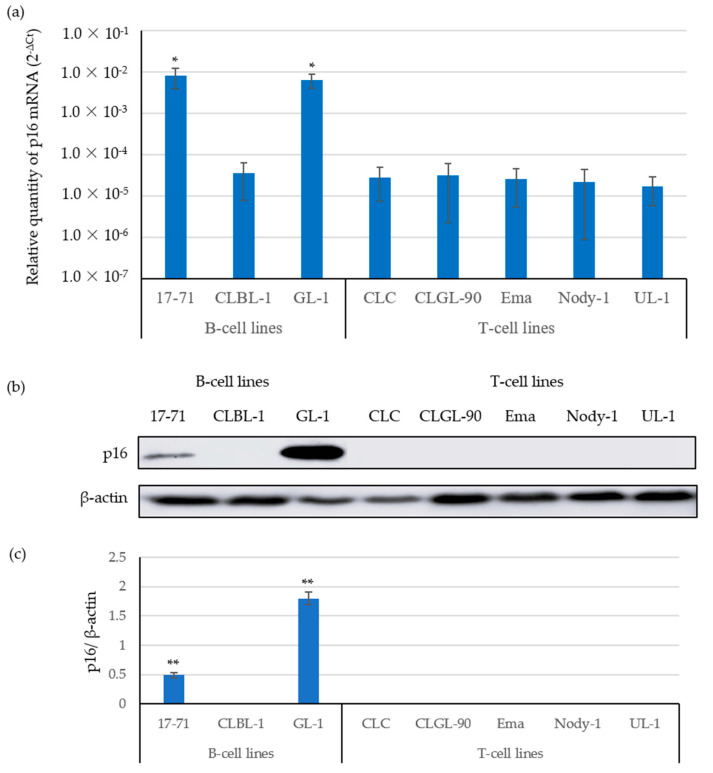
The expression analysis of the p16 gene and protein in canine lymphoma B- and T-cell lines. The original images were provided as Appendix A. (**a**) The relative expression levels of p16 mRNA were assessed using real-time PCR. The data are expressed as the mean and standard deviation (SD) values of three replicates in the triplicate assay. * *p* < 0.05. (**b**) The expression analysis of the p16 protein in canine lymphoma cell lines. The protein expression was assessed using western blot analysis (**b**) and quantified using ImageJ (**c**). ** *p* < 0.01.

**Figure 2 vetsci-09-00393-f002:**
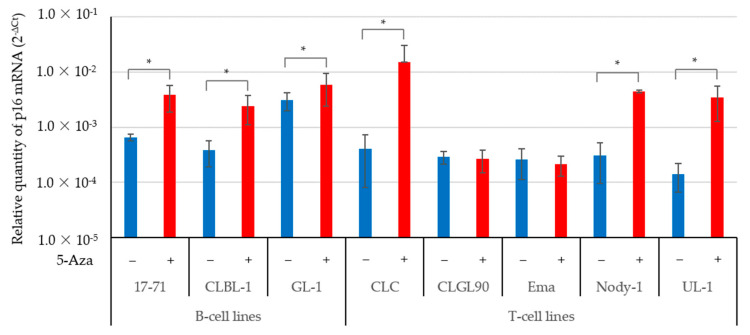
The expression of the p16 gene in canine lymphoma and leukemia B- and T-cell lines treated with 5-Aza. The relative expression levels of p16 mRNA in the 17-71, CLBL-1, and GL-1 B-cell lines; and the CLC, Nody-1, and UL-1 T-cell lines were significantly increased after the 5-Aza treatment, as assessed using real-time PCR. The data are expressed as the mean and SDs values of three replicates in the triplicate assay. * *p* < 0.01.

**Figure 3 vetsci-09-00393-f003:**
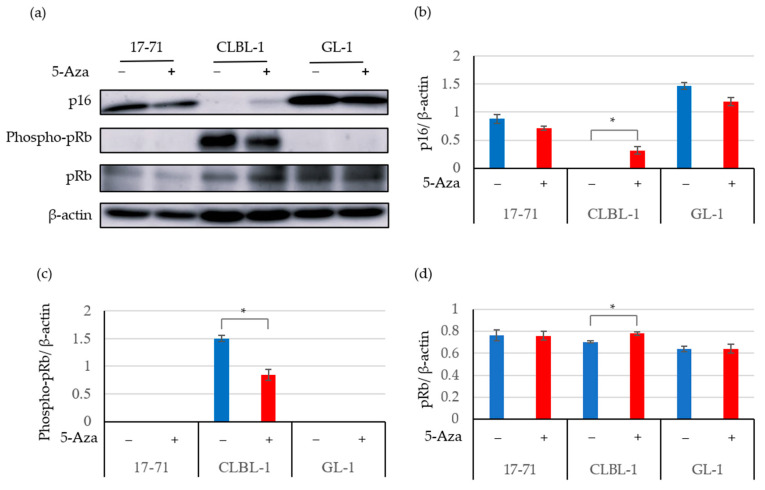
The expression of the p16 protein in canine lymphoma and leukemia B-cell lines treated with or without 5-Aza. The original images were provided as Appendix A. The protein expression was assessed using the western blot analysis (**a**) and quantified using ImageJ for p16 (**b**), phospho-pRb (**c**), and total pRb (**d**). β-actin was used as the endogenous control. The data are expressed as the mean and SD values. * *p* < 0.01.

**Figure 4 vetsci-09-00393-f004:**
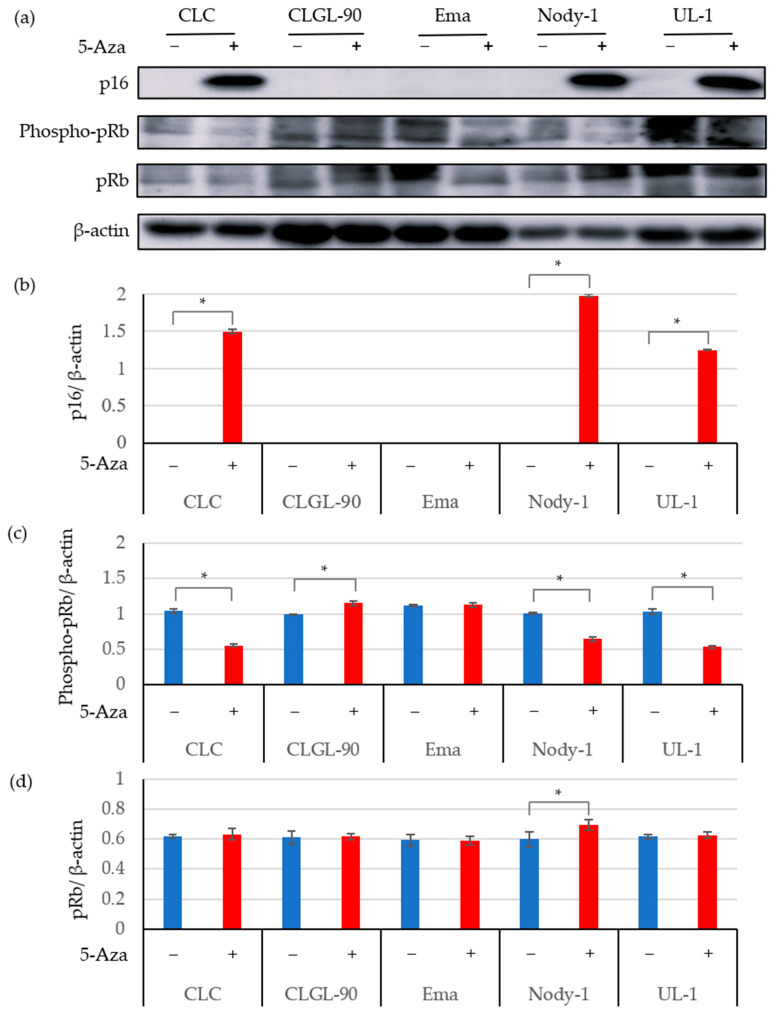
The expression analysis of the p16 protein in canine lymphoma T-cell lines treated with or without 5-Aza. The original images were provided as Appendix A. The protein expression was assessed using the western blot analysis (**a**) and quantified using ImageJ for p16 (**b**), phospho-pRb (**c**), and total pRb (**d**). β-actin was used as the endogenous control. The data are expressed as the mean and SD values. * *p* < 0.01.

**Table 1 vetsci-09-00393-t001:** Primer sequences used for canine p16 and RPL32 genes.

Target Gene	Primer Name	Sequence (5′-3′)	Product Length (bp)	GeneBank Accession No.
p16	16F	GGTCGGAGCCCGATTCA	95	AB675384
	16R	ACGGGGTCGGCACAGTT		
RPL32	RPL32F	TGGTTACAGGAGCAACAAGAA	100	XM848016
	RPL32R	GCACATCAGCAGCACTTCA		

**Table 2 vetsci-09-00393-t002:** Antibodies used for the detection of canine p16, phospho-pRb, pRb and β-actin proteins.

Target Protein	Clone	Manufacture	Diluted with	Dilution
p16	F-8	Santa Cruz Biotechnology	5% nonfat milk/TBST	1:500
phospho-pRb	Phospho-T826	Abcam	5% nonfat milk/TBST	1:1000
pRb	G3-248	BD Pharmingen	5% nonfat milk/TBST	1:1000
β-actin	AC-15	Sigma–Aldrich	0.5% nonfat milk/TBST	1:5000

**Table 3 vetsci-09-00393-t003:** Summary of the expression analysis of the p16 gene and protein, and the pRb phosphorylation in canine lymphoma cell lines treated with (+) or without 5-Aza (−).

Canine Lymphoma Cell Lines		p16					Phospho-pRb		
		Gene Expression		Protein Expression		Methylation	Protein Expression		Hyper-Phosphorylation
		5-Aza					5-Aza		
		(−)	(+)	(−)	(+)		(−)	(+)	
B-cell lines	17-71	++	+++	++	++	−	−	−	−
	CLBL-1 *	++	+++	−	+	+	++++	++	+
	GL-1	+++	++++	+++	+++	−	−	-	−
T-cell lines	CLC *	+	++++	−	+++	+	++	+	+
	CLGL-90	+	+	−	−	−	++	+++	+
	Ema	+	+	−	−	−	++	++	+
	Nody-1 *	+	+++	−	+++	+	++	+	+
	UL-1 *	+	+++	−	+++	+	++	+	+

* The cell lines in the grey column showed low expression of the p16 gene and protein, in which the methylation of p16 gene and the phosphorylation of pRb might have occurred.

## Data Availability

All data is presented in this study, and further inquiries can be directed to the corresponding author.

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
