# Peer review of "Simultaneous Analysis of the p16 Gene and Protein in Canine Lymphoma Cells and Their Correlation with pRb Phosphorylation"

_vetsci, 2022, doi:10.3390/vetsci9080393_

Round 1

Reviewer 1 Report

In this paper, the authors present their work about analysis of the p16 gene and protein in canine lymphoma cell lines and their correlation with pRb.

Following are comments regarding the manuscript:

Lines 20-21 and line 24: please use full cell line names in the Abstarct

Line 47: why there is 80% in bracets?

Line 92: how the cell colonies was optimized?

Line 110: "...determinated based on our preliminary experiments." Can you be more specific or maybe could you provide reference or briefly describe why such dosage was used?

Line 115: how the RNA quantity was obtained? Whether RNA standarization was made before transcription into cDNA? What was the control group in PCR?

Line 203: in Figure 1b why there isn't control?

Lines 435, 442, 446: why do you cite figures one more time?

Lines 435-439: I would move that infromation to the Results section.

Lines 451-460: remove the whole paragraph.

Reviewer 2 Report

Reviewer´s comments to the authors:

A brief summary:

The article entitled “Simultaneous Analysis of the p16 Gene and Protein in Canine Lymphoma Cells and their Correlation with pRb Phosphorylation” by Maylina et al. aims to provide a better understanding of the relationship of the gene and protein expression of p16 with phosphorylated pRb, both involved in the cell cycle. For this in vitro study, the authors use 8 canine lymphoma and leukemia cell lines, and they determine the p16 gene and protein expression, with or without methylation of this gene, and assessed their association with phosphorylation of the RB protein.  The current study is part of an investigation focused on the treatment of canine lymphoma.

The innovative contribution of this study is the examination of p16 protein expression, which has not been assessed in previous studies, in its relationship with pRB activation, taking into account the inactivation or not of the p16 gene.

General concept comments:

Strengths of this study

·         This in vitro study is relevant to support the role of the p16 and retinoblastoma protein as an important pathway, but not the only, in the development of canine lymphomas.

·         The novelty with respect to previous studies is the analysis of the p16 protein.

·         The results provide an advance in current knowledge of the canine lymphoma cells for future research into treatment of this disease.

Weaknesses of this study

·         The mechanism of p16 gene hypermethylation, decreased p16 protein expression, and hyperphosphorylated pRB seemed to be involved in canine lymphomagenesis in only 4 out of 8 canine lymphoma cell lines.

·         These results suggest that other canine lymphoma and leukemia cell lines may have other mechanisms.

·         The results may be biased by the management of the preservation and culture of the cell lines in laboratories.

Specific comments:

·         The experimental design, the methods used (demethylation treatment of cell lymphoma cell lines, PCR and Western blot), and the parameters assessed (p16 gene and protein expression, and phosphorylation of the RB protein) are appropriate.

·         The manuscript is well structured.

·         The result section is easy to read and understand thanks to the tables and figures the authors included.

·         The discussion section is well written, relating the results with information from the literature.

·         The conclusions answer the aims of the study.

·         Bibliographic references are adequate, and they include three articles published in 1988, 1996 and 1998. The remaining articles were published between 2003 and 2021.

This article is an interesting study but needed minimal revision for the final version.

Abstract section

Line 19. “in vitro study” should be added in abstract section.

 Materials and Methods section

Line 110. In the sentence on the doses used for treatment of the canine lymphoma cell lines with 5-aza-2-deoxycytidine, the authors should include the bibliographical references of the “preliminary experiments”.

Lines 156-162. In the “Statistical methods” section, the authors indicate that one-way ANOVA and Tukey´s post-test were used. These parametric tests are carried out when the variables follow a normal distribution. Was normality tested?

Discussion section:

Line 372. The punctation mark “;” should be deleted in the sentence “…the loss of the p16 (CDKN2A) and TP53 genes was observed………”.

Line 391. The punctation mark “;” should be deleted in the sentence “…and GL-1 cells were not related……

Conclusions:

Line 391. The authors should add the word “more” in the following sentence: “….because the expression of the protein was more stable tan those of the gene,…..”

Round 2

Reviewer 1 Report

Please accept the manuscript in present form.